# The Impact of Language Variability on Artificial Intelligence Performance in Regenerative Endodontics

**DOI:** 10.3390/healthcare13101190

**Published:** 2025-05-20

**Authors:** Hatice Büyüközer Özkan, Tülin Doğan Çankaya, Türkay Kölüş

**Affiliations:** 1Department of Endodontics, Faculty of Dentistry, Alanya Alaaddin Keykubat University, 07490 Alanya, Türkiye; tdogancankaya@gmail.com; 2Department of Restorative Dentistry, Faculty of Dentistry, Karamanoğlu Mehmetbey University, 70200 Karaman, Türkiye; turkaykolus@hotmail.com

**Keywords:** artificial intelligence, regenerative endodontic procedures, ChatGPT, Claude, Grok, Gemini, endodontics, dental education

## Abstract

Background: Regenerative endodontic procedures (REPs) are promising treatments for immature teeth with necrotic pulp. Artificial intelligence (AI) is increasingly used in dentistry; thus, this study evaluates the reliability of AI-generated information on REPs, comparing four AI models against clinical guidelines. Methods: ChatGPT-4o, Claude 3.5 Sonnet, Grok 2, and Gemini 2.0 Advanced were tested with 20 REP-related questions from the ESE/AAE guidelines and expert consensus. Questions were posed in Turkish and English, with or without prompts. Two specialists assessed 640 AI-generated answers via a four-point rubric. Inter-rater reliability and response accuracy were statistically analyzed. Results: Inter-rater reliability was high (0.85–0.97). ChatGPT-4o showed higher accuracy with English prompts (*p* < 0.05). Claude was more accurate than Grok in the Turkish (nonprompted) and English (prompted) conditions (*p* < 0.05). No model reached ≥80% accuracy. Claude (English, prompted) scored highest; Grok-Turkish (nonprompted) scored lowest. Conclusions: The performance of AI models varies significantly across languages. English queries yield higher accuracy. While AI shows potential for REPs information, current models lack sufficient accuracy for clinical reliance. Cautious interpretation and validation against guidelines are essential. Further research is needed to enhance AI performance in specialized dental fields.

## 1. Introduction

Artificial intelligence (AI) has emerged as a transformative technology across numerous industries, including healthcare and dentistry. The integration of AI in medical practice has roots in mid-20th-century experiments involving computer-assisted data interpretation and diagnostic frameworks [1]. Over the past year, these foundational systems have rapidly matured into complex architectures capable of clinical decision making with reliability metrics approaching those of human performance. Current healthcare AI deployments demonstrate practical potential in various domains, such as automated radiological pattern recognition, genomic-data-based therapy design, real-time vital sign analytics, and probabilistic modeling of pathological progression. These models show promising results in medical and dental applications and can potentially improve diagnosis, treatment planning, and patient care [2,3,4]. In dentistry, particularly in endodontics, AI has already shown promise in various applications, such as analyzing root canal anatomy, detecting periapical lesions, and predicting treatment outcomes [5,6].

Large language models (LLMs) are a form of AI using deep learning techniques to understand, summarize, and generate textual content [7]. Recent research demonstrates the capacity of these systems to process and analyze vast quantities of medical literature, thereby offering dental professionals quick access to up-to-date information [8]. These developments suggest that AI and LLMs could play a significant role in disseminating knowledge about regenerative endodontic procedures (REPs).

REPs have emerged as promising treatment modalities for immature permanent teeth with necrotic pulp, allowing the regeneration of the pulp–dentin complex, facilitating continued root development, and preserving natural teeth that would otherwise be at risk of extraction [9]. The limitations of conventional endodontic treatments, such as apexification, which does not promote root development or apical closure, drive the need for REPs. Although comprehensive epidemiological data for monitoring the requirements for REPs are lacking in the literature, conditions such as caries, dental abnormalities, and traumatic dental injuries lead to pulp necrosis, where REPs may be beneficial. Dental trauma is common across all ages, with a particularly high prevalence in childhood. One European study revealed that 15% of preschoolers and 20–25% of school-age children experience dental trauma [10]. The prevalence of pulp necrosis following traumatic dental injuries varies substantially, ranging from 17 to 100%, depending on the severity of the injury [11]. In a study of adolescent populations, the overall prevalence of traumatic dental injuries was 16.4%, with 7.5% of these patients developing pulp necrosis with infection [12]. The occurrence of pulpal necrosis in permanent but immature teeth presents a challenging clinical situation for which REPs can offer significant benefits in healing apical periodontitis, promoting continued root development, and restoring functional pulpal tissue.

Recent studies indicate that REPs have achieved high success rates, ranging from 83.3–100% in clinical applications [9]. Additionally, the survival rates for teeth treated with REPs range from 94 to 100%, with a 5-year follow-up study showing a sustained 92% tooth survival rate [13]. A comprehensive meta-analysis revealed that the success rates for necrotic immature and mature permanent teeth treated with REPs were 95.6% and 95.5%, respectively, demonstrating the effectiveness of the procedure across different tooth development stages [14]. These statistics underscore the growing importance of REPs in modern endodontic practice. Despite promising outcomes, dentists face several significant challenges when performing REPs, such as unpredictable outcomes, a lack of standardized protocols, complex root canal anatomy, infection control challenges, coronal sealing issues, root resorption risks, and crown discolorations. As this relatively novel approach has gained traction in clinical practice, there is an increasing need for accurate and accessible information to guide dental practitioners in its application. Dentists often face uncertainty when selecting appropriate cases, managing complications, or interpreting evolving guidelines, which can lead to inconsistent clinical results. The need for reliable, up-to-date, and easily accessible information is therefore critical for improving the predictability and safety of REPs. Providing dentists with comprehensive, reliable information on REPs through accessible formats, including text, images, and videos, would offer several advantages, such as improved clinical decision making, standardization of procedures, enhanced training opportunities, complication management, and patient education.

The potential of AI to provide quick and accessible information could significantly impact the adoption and successful implementation of these procedures in clinical practice. However, owing to the inherent limitations of LLMs, such as the potential for inaccuracies, biases, and outdated content, along with the complexity and evolving nature of REPs, critically examining AI-generated information is crucial [15].

The intersection of AI and REPs presents an intriguing research area, as AI systems could serve as a valuable resource for dentists seeking information on this evolving treatment modality. However, AI-generated information on REPs remains a critical concern that warrants a thorough investigation [6]. Some studies have examined the reliability of AI-generated content about REPs [16,17]. However, the responses of more than two AIs to questions related to REPs in two different languages have not been compared. Turkish (Tr) was chosen as the primary language for this study, as the questions were designed and administered in Turkey to facilitate better comprehension among local practitioners. However, this choice raises concerns about the potential loss of meaning or accuracy in AI processing, highlighting a critical knowledge gap regarding the impact of linguistic variability on AI performance. Given the potential for language-specific nuances and the lack of comprehensive research on AI performance across different languages in dental contexts, this study includes questions in both Tr and English (Eng).

Given their popularity and extensive evaluation in the recent scientific literature, four state-of-the-art AI models were selected for this study. These LLMs were chosen because of their broad accessibility, wide usage, and general-purpose functionality, making them more representative of the models dental practitioners and students may use in real-world scenarios when seeking information. ChatGPT-4o, developed by OpenAI, is widely recognized for its conversational fluency, adaptability, and broad applicability across text generation, knowledge retrieval, and customer support domains. Grok 2, designed by xAI, integrates real-time social media data, particularly from the X (formerly Twitter) platform, which enhances its ability to analyze trends, generate insights, and engage in context-aware discussions. Gemini 2.0 Advanced, developed by Google DeepMind, excels in multimodal AI, processing text, images, and video, and is highly integrated into Google’s ecosystem, thereby enhancing its utility for AI-driven decision-making tasks [18]. Claude 3.5 Sonnet, developed by Anthropic, is an LLM designed with a particular emphasis on ethical considerations and safety, incorporating a more controlled and transparent approach to AI interactions [19].

This study aimed to evaluate and compare the responses provided to questions related to REPs by state-of-the-art AI models, including ChatGPT-4o (GPT), Claude 3.5 Sonnet (Claude), Grok 2 (Grok), and Gemini 2.0 Advanced (Gemini), against the clinical guidelines established by the European Society of Endodontology (ESE) [20] and the American Association of Endodontists (AAE) [21] and the expert consensus published in 2022 [9].

The first H_0_ was that there would be no significant difference in the responses provided by each AI model when presented with questions related to REPs under four different conditions: prompted, nonprompted, in Tr, and in Eng. The second H_0_ was that there would be no significant difference in the responses to REP-related questions among different AI models.

## 2. Materials and Methods

This research did not require ethical approval since no human participants were involved.

### 2.1. Question Design

To ensure a systematic and clinically relevant evaluation of AI-generated information, we developed 20 questions according to the most current and authoritative sources in regenerative endodontics, including the European Society of Endodontology (ESE) Position Statement, the American Association of Endodontists (AAE) Clinical Considerations for a Regenerative Procedure, and the 2022 expert consensus. According to the power analysis conducted via GPower (Universität Kiel, Kiel, Germany, version 3.1.9.4 software), when the effect size was assumed to be 0.75, the required sample size per group to achieve 80% power was calculated as four observations. Since the current study included 20 observations per group, the actual statistical power was determined to be 0.99999, indicating an extremely robust analysis.

Two board-certified endodontists independently reviewed the questions to ensure content validity, clinical relevance, and alignment with guideline-based practices. Both evaluators were certified endodontic specialists, officially recognized by the Republic of Türkiye Ministry of Health, with over 10 years of combined clinical and academic experience. These experts are actively involved in postgraduate dental education and possess comprehensive theoretical and practical expertise in regenerative endodontic procedures. This expert-based review process provided a robust foundation for the methodological integrity of the question set.

This study compared responses from four AI models (Gemini 2.0 by Google, GPT 4o by OpenAI, Claude 3.5 Sonnet by Anthropic, and Grok 2 by xAI) to questions about REPs. Twenty questions were developed on the basis of three sources: ESE position statements [20], the AAE “Clinical Considerations for a Regenerative Procedure” [21], and the Expert Consensus on Regenerative Endodontic Procedures [9]. The designed questions are listed in Table 1.

### 2.2. Generating Answers

Answers were generated for each question via the ‘new chat’ function in four different AI models. The questions were posed to each AI at 10-day intervals between 1 January and 15 January 2025, to assess repeatability; all the responses were recorded. Each question was administered in both Eng and Tr, with each language group divided into two subgroups: prompted and nonprompted. In the prompted subgroups, standardized instructions were provided—“Respond as an endodontic specialist” for English and “Bir endodonti uzmanı olarak yanıt ver” for Turkish. No additional instructions were given in the nonprompted subgroups. This design resulted in four experimental conditions:Eng promptEng nonpromptTr promptTr nonprompt

A total of 640 answers were collected and compiled into a Microsoft Word document (Version 16.93, Microsoft, Redmond, WA, USA).

### 2.3. Evaluation of AI Answers by Human Experts

Two board-certified endodontists independently assessed all AI-generated answers via a 4-point holistic scoring rubric to evaluate response consistency. Before scoring, a calibration session was conducted between the two evaluators to standardize the evaluation criteria according to ESE and AAE guidelines and the 2022 expert consensus document. This process ensured methodological consistency and strengthened inter-rater reliability.

The evaluation criteria were applied as follows:

1 = Incorrect (the AI’s answer is completely wrong).

2 = Partially correct (the AI’s answer contains both correct and incorrect information).

3 = Correct but incomplete (the AI’s answer is correct but incomplete).

4 = Fully correct (the AI’s answer is completely correct).

### 2.4. Statistical Analysis

The data were analyzed via the IBM SPSS Statistics V24.0 (IBM Corp, Armonk, NY, USA) statistical package program. The analysis results were considered significant at the 95% confidence level with *p* values below 0.05.

The consistency between the two expert raters’ scores was calculated via the Brennan–Prediger coefficient, Cohen’s kappa coefficient, Fleiss’s kappa coefficient, and Krippendorff’s alpha coefficient. Temporal consistency and test–retest reliability were evaluated via the intraclass correlation coefficient (ICC) and Pearson correlation coefficients.

The normality assumption using the Kolmogorov–Smirnov and Shapiro–Wilk tests was assessed to determine the appropriate hypothesis testing methodology. These tests confirmed a normal data distribution (*p* > 0.05), justifying the use of parametric statistical techniques. Within the scope of descriptive statistics, the total mean and standard deviation of each AI model under different prompting conditions were calculated and are shown in Table 2.

One-way analysis of variance (ANOVA) was conducted to assess significant differences in mean response scores across AI models (GPT, Claude, Grok, and Gemini) for the Eng and Tr languages under prompted and nonprompted conditions. Additionally, the average responses of different AI models were compared, and differences between the models were analyzed for significance. Pairwise comparisons between AI models were performed via Tukey’s HSD post-hoc test to identify specific group differences when ANOVA revealed significant results.

## 3. Results

Table 3 presents the repeatability and agreement coefficients between the two expert raters’ scores. The Brennan–Prediger, Cohen’s kappa, Fleiss’ kappa, and Krippendorff’s alpha coefficients for inter-rater reliability were calculated as 0.97, 0.88, 0.86, and 0.85, respectively. The ICC and Pearson correlation coefficient for temporal consistency and test–retest reliability were 0.92 and 0.89, respectively. Given these results and strong consistency, the scores of the AI-generated responses were computed by averaging the evaluations from both endodontists.

When the AI models’ responses to both the Tr and Eng questions (prompt and nonprompt) were evaluated, one-way ANOVA revealed that, compared with the Tr models, only the GPT model exhibited significantly greater accuracy in Eng-prompt responses (*p* < 0.05). Table 4 shows that the differences between the groups (sum of squares = 1.834, df = 3) are significant compared with the differences within the groups (sum of squares = 55.888, df = 76), with an F value of 1.832 and a *p* value of 0.033. Since the *p* value is below 0.05, the result indicates a significant difference between the groups. For Claude, Grok, and Gemini, no statistically significant differences were detected between the Tr and Eng responses across the prompting conditions (*p* > 0.05).

Pairwise comparisons revealed no significant differences between GPT and the other tested AI models (*p* > 0.05). Similarly, no significant differences were detected between Claude and Gemini (*p* > 0.05) or between Grok and Gemini (*p* > 0.05). However, a significant difference was identified in the Claude vs. Grok comparison (*p* < 0.05). In the Tr nonprompt assessment, Claude demonstrated significantly greater response accuracy than Grok did. Similarly, in the Eng prompt group, Claude’s accuracy was significantly superior to that of Grok’s.

Table 5 shows the results of a one-way ANOVA comparing response accuracy across Grok and Claude under language-prompting conditions and the group differences across Tr nonprompt, Eng prompt, Tr prompt, and Eng nonprompt. Significant differences were found in the Tr nonprompt (F = 1.466, *p* = 0.041) and Eng prompt (F = 1.390, *p* = 0.046) conditions, where post-hoc analysis indicated that Claude performed better than Grok did. No significant group differences were detected in the Tr prompt (F = 0.903, *p* = 0.444) or Eng prompt (F = 1.277, *p* = 0.288) conditions, as indicated by *p* values above 0.05. Therefore, group performance differences were significant only in the Tr nonprompt and Eng prompt conditions.

Figure 1 shows the number of fully correct responses generated by four AI models—GPT, Claude, Grok, and Gemini—across four evaluation conditions: Tr prompt, Tr nonprompt, Eng prompt, and Eng nonprompt. Claude achieved the highest score in the Tr prompt setting with nine fully correct responses, followed by Gemini with seven, GPT with five, and Grok with four. For the Tr nonprompt condition, both Claude and Gemini performed equally well, each providing seven correct responses, whereas GPT produced six and Grok produced only two. In the Eng prompt condition, both GPT and Claude scored ten fully correct responses, Gemini achieved nine, and Grok managed five. Finally, in the Eng nonprompt condition, Claude, GPT, and Gemini each recorded seven correct responses, whereas Grok lagged by only three. Overall, Claude consistently demonstrated strong performance across all conditions, whereas Grok presented the lowest scores, particularly in the nonprompt tasks.

Figure 2 shows the mean scores of AI-generated responses across four conditions: Tr prompt, Tr nonprompt, Eng prompt, and Eng nonprompt. The performances of four AI models—GPT, Claude, Grok, and Gemini—are compared for each condition. Claude consistently achieved the highest mean scores in all conditions, with its performance peaking in the Eng prompt scenario (mean ≈ 3.225). GPT and Gemini followed closely behind Claude, whereas Grok consistently received the lowest mean scores across all conditions. The differences between the models were most pronounced in the Tr nonprompt and Eng prompt conditions, where Claude’s lead over Grok was especially notable. Overall, the chart highlights Claude’s superior performance and Grok’s relatively lower scores in generating responses, regardless of language or prompt presence.

## 4. Discussion

The first H_0_ was rejected because significant differences were observed in the responses provided by each AI model when questions about REPs were presented under different conditions (prompted, nonprompted, in Tr, and in Eng). The second H_0_ was also rejected because our analysis revealed substantial variations in the different AI models’ responses to REP-related questions.

LLMs are deep learning models designed to comprehend and generate meaningful responses via a multilayer neural network architecture [22]. LLMs are trained on extensive volumes of textual data in an unsupervised fashion, enabling them to make inferences about the relationships between words in the text. In addition, these models can predict the next word in a sequence of words given the previous words. AI frameworks consist of different modeling approaches, including the generative model (GM), retrieval model (RM), and retrieval-augmented generation (RAG). The GM uses neural networks to generate text according to patterns learned from extensive datasets. This approach can produce novel responses and address queries with remarkable fluency. Moreover, it can generate consistent and creative outputs through statistical analyses. However, this creativity can sometimes result in the generation of inaccurate, misleading, or fabricated information, a phenomenon known as “hallucination” [23]. In clinical contexts, such hallucinations pose a significant risk, as they may lead to erroneous decision making or dissemination of incorrect clinical advice. Therefore, despite the promising capabilities of AI models, human verification remains indispensable, particularly in healthcare settings where accuracy is critical [24].

The use of AI models is advancing in various fields, including dentistry. AI models generate words and sentences according to probability distributions using web information as learning data. Owing to this probabilistic nature, different responses may be generated when the same question is asked multiple times to the same or different AI model [25].

Recent advancements in AI have led to the development of several sophisticated language models, each with unique capabilities and applications. GPT-4o, launched on 14 May 2024, and developed by OpenAI, demonstrates enhanced multimodal processing abilities, integrating text, audio, and visual inputs for improved reasoning and real-time interactions [26]. GPT-4o offers a large context window (128K tokens) and is accessible via the web and API, with both free and paid tiers. Limitations include a knowledge cutoff (October 2023), potential hallucinations, and message limits for free users. Anthropic’s Claude 3.5 Sonnet, released on 21 June 2024, showcases significant improvements in coding, mathematics, and visual understanding with a 200,000-token context window and a transformer architecture optimized for natural language processing, focusing on ethical alignment, advanced reasoning, and coding capabilities. Access is provided through commercial APIs. Its main limitations are the lack of real-time web access and proprietary training data. Grok 2, launched on 13 August 2024, created by xAI, incorporates advanced natural language processing and image generation capabilities, emphasizing user engagement and real-time information access. While Grok 2 excels at providing up-to-date information, it has limited adoption, potential data privacy concerns, and is heavily dependent on the X platform. Its developer ecosystem and third-party integrations are still maturing. Finally, Google DeepMind’s Gemini 2.0 Advanced was released on 11 December 2024, and exhibits enhanced multimodal understanding, improved reasoning, and problem-solving abilities across various domains. These models represent the cutting edge of AI technology, each offering unique features that have the potential to revolutionize how we interact with and utilize AI in various fields, including academic research and practical applications [27].

Dentists may prefer to access information in their native language for ease of comprehension; thus, we evaluated AI responses to questions posed in both Tr and Eng. When assessing the accuracy of GPT responses, we observed significant improvements in Eng-prompted answers compared with Tr-prompt ones. Tr possesses an agglutinative linguistic structure [28] and presents distinctive challenges for natural language processing systems because of its complex morphological characteristics [29]. This complexity has been reported to significantly impact the performance of language models, with agglutinative languages consistently showing lower performance measures than fusional languages do [29]. Another reason may be that the marker methods used by GPT models, such as byte-pair coding, are not suitable for the morphological structure of Tr [30]. These tokenization methods are generally optimized for Eng and may not be able to effectively handle the rich morphology of Tr. Additionally, these models are predominantly trained on Eng-centric corpora, leaving Tr-specific tokens underrepresented during the training phase. Although other AI models (Claude and Gemini) yielded higher mean accuracy scores for Eng responses, these differences were not significant (Table 3). Only Grok’s Tr prompted average score was higher than Grok’s other scores. Previous research emphasizes the importance of question language and sources when validating chatbot responses [31]. However, no prior study systematically compares the consistency of answers generated by AI models across different languages. Our work addresses this gap, highlighting that while only GPT exhibited significant language-dependent variation, GPT, Claude, and Gemini demonstrated numerically greater accuracy in Eng, underscoring the critical role of language selection in AI-powered dental applications.

In the present study, Grok provided more general and shorter answers in both languages, whether prompted or not. In addition, Grok had the lowest accuracy level among the AI models. Especially in the Tr nonprompt condition (62.50% accuracy rate), Grok 2 had the lowest average. Grok is known to be trained on real-time data from the X (Twitter) platform; however, these data may be noisy and contain misinformation or lack cultural context. The limited availability of high-quality and balanced datasets, especially for low-resource languages such as Tr, may reduce the model’s accuracy [32]. The differences in chatbot responses can also be attributed to the varying design philosophies of the AI companies, the specific algorithms employed, the datasets used for training, and the objectives the AI is designed to achieve [33]. The performance difference between language models may be due to the size differences in the training datasets [29]. The volume of digital data available for Tr is quite limited compared with that available for Eng [34]. These reasons may explain why Grok was the lowest performing AI model in our study; however, more detailed studies are needed.

In this study, the evaluation extended beyond the accuracy rates of AI models to include a qualitative analysis of the types of content errors observed in their responses. These errors were thematically categorized into three main groups:(i)Minor inaccuracies with low theoretical significance (e.g., vague definitions);(ii)Moderate errors or omissions that may indirectly affect clinical procedures (e.g., anesthetic selection or irrigation sequence);(iii)Clinically significant misinformation that conflicts with established guidelines.

Notably, both Claude and GPT occasionally recommend chlorhexidine, which is contraindicated in REPs because of its known cytotoxicity to apical tissues. Additionally, GPT’s recommendation to use sodium hypochlorite during the second appointment was inconsistent with current clinical protocols. Such misinformation may be particularly misleading for early-career dentists or dental students, as AI-generated responses can appear authoritative and evidence-based. Although Claude and GPT demonstrated relatively high accuracy scores overall, they were still prone to providing outdated or guideline-inconsistent suggestions. These findings underscore that strong performance metrics do not guarantee error-free outputs and highlight the necessity for expert oversight when AI tools are utilized in clinical dentistry.

In this study, although no direct misinterpretation of REP-related terminology was observed in the Tr responses, some clinical concepts were conveyed with reduced precision, particularly in nonprompted conditions. These discrepancies did not stem from the mistranslation of specific terms but from limitations in contextual understanding and procedural clarity. In some cases, AI responses lacked the depth or specificity needed to fully align with clinical guidelines despite using technically correct vocabulary. This result may be partly explained by the linguistic structure of Tr. As an agglutinative language, Tr expresses grammatical and semantic relationships through complex suffixation patterns, which can pose challenges for AI models primarily trained on fusional languages such as Eng. In some Tr outputs, the main action or verb in the question was occasionally misinterpreted or underemphasized, potentially leading to less accurate or incomplete responses. These findings underscore the importance of optimizing language models for morphologically rich languages to improve the reliability of AI-generated clinical content across multilingual settings.

Díaz-Flores García et al. [17] conducted a comprehensive study to evaluate Google Gemini’s performance in endodontics, focusing on the accuracy and reliability of its answers to diagnostic and treatment questions. The researchers designed 60 questions on the basis of the European Society of Endodontology Position Statements; 30 questions were randomly selected for the study, of which 9 were specifically related to REPs. They generated 900 answers via Gemini in April 2023, which were then independently scored by two endodontic experts via a three-point Likert scale. The study revealed that Gemini’s overall accuracy was 37.11% (95% confidence interval: 34.02–40.32%). In our study, Gemini 2.0. The prompt accuracy was 76.25%. The Gemini model tested in our study was a newer version; thus, our findings may be an important demonstration of the development of the AI model over time. The researchers observed high repeatability in Gemini’s responses but noted that this consistency also extended to incorrect answers; in our study, Gemini 2.0 also showed high repeatability and consistency. They concluded that while Gemini shows potential as a diagnostic support tool, its current performance may not be sufficiently reliable for critical clinical applications in endodontics, emphasizing the need for caution when considering its use in medical settings.

A study conducted by Ekmekci and Durmazpınar [16] evaluated three AI platforms—Google Bard (Gemini), ChatGPT-4o, and ChatGPT-4 with the PDF plugin—for their ability to answer clinical questions about regenerative endodontic treatment. Using 23 questions derived from the AAE’s 2022 guidelines, researchers collected 1380 responses over 10 days through three daily sessions. The PDF-equipped ChatGPT-4 demonstrated superior performance, with 98.1% accuracy in clinical recommendations, no incorrect responses, and only 1.9% insufficient answers. This approach significantly outperformed both ChatGPT-4o (86.2% correct) and Gemini (48% correct), which presented higher rates of insufficient (10.9–34.5%) and incorrect (2.9–17.5%) responses. Statistical analysis revealed significant differences in response accuracy across platforms (*p* < 0.05), particularly concerning Gemini’s inconsistent performance compared with ChatGPT variants. This study highlights ChatGPT-4’s PDF analysis capability as a particularly effective tool for retrieving guideline-based endodontic information.

Özden et al. [27] asked ChatGPT and Google Bard 25 yes/no questions prepared according to the International Association of Dental Traumatology guidelines over 10 days, comparing the AI models’ responses with correct answers. The study revealed that while ChatGPT and Google Bard represent potential information sources, their response consistency and accuracy regarding dental trauma queries have limitations. The researchers emphasized the need for further research on AI models specifically trained in endodontics to assess their clinical applicability. Our findings align with these results.

Suarez et al. [35] evaluated ChatGPT’s clinical decision-making capabilities by posing 60 dichotomous (yes/no) questions in line with the guidelines of the European Society of Endodontics (ESE) of varying difficulty levels (20 easy, 20 medium, 20 difficult) to ChatGPT-4. Two independent endodontists validated the AI responses. The researchers opted for a yes/no format to simplify direct comparisons between ChatGPT’s outputs and expert evaluations. Their results revealed 85.44% response consistency (no significant variation across difficulty levels) but only 57.33% accuracy compared with expert judgments. Notably, ChatGPT performed worse on easier questions (50.83% accuracy) than on difficult ones (63.33%), suggesting gaps in foundational clinical knowledge. The study concluded that while ChatGPT cannot currently replace dentists, its utility in endodontics may improve with deep learning advancements. However, the authors emphasized stringent validation protocols to ensure AI system reliability, citing the risk of “hallucinations” (plausible but incorrect responses) due to training data limitations (pre-2021 cutoff) and a lack of medical database access. In the current study, ChatGPT-4o, a more advanced iteration of the model, demonstrated improved accuracy (78%) in endodontic clinical decision-making tasks. This finding aligns with Suarez et al.’s on AI’s evolving potential while reinforcing their call for rigorous validation processes, particularly as model performance advances. Both studies underscore that even higher accuracy rates necessitate continuous evaluation to address persistent challenges such as temporal data gaps and contextual comprehension limitations.

Mohammad-Rahimi et al. [31] asked GPT-3.5, Google Bard, and Bing 20 questions about endodontics and root canal treatments. They stated that while GPT-3.5 yielded promising results regarding the validity of its responses, there are areas for future improvements for all tested AI models. Additionally, as a result of this research, the authors stated that as chatbot technologies advance, it is highly important for researchers and endodontic societies/associations to evaluate their performance and share their limitations with the public to inform the public about valid information and potential misinformation provided by AI chatbots.

The importance of this research lies in the potential impact of AI as an information source for dental practitioners seeking knowledge about REPs. As a relatively new treatment modality, REPs require careful consideration and understanding of the latest evidence-based protocols.

Furthermore, this study contributes to the broader discussion on the role of AI in dental education and clinical practice. As AI technologies continue to evolve, critically assessing their performance in providing accurate and clinically relevant information, particularly for emerging treatments such as REPs, is crucial [36,37].

The scientific contribution of this work is that it is the first systematic evaluation of how language variability, specifically, differences between Tr and Eng, affects the accuracy and reliability of AI models in providing information about REPs. Our study reveals that AI-generated responses about REPs show significant performance variations depending on the language and prompting strategy used. Notably, Eng prompts consistently yielded higher accuracy than Tr, highlighting a critical gap in AI model performance for methodologically complex, specialized languages. This research not only identifies the current limitations of AI models in delivering clinically accurate information across languages but also provides a robust methodological framework for future benchmarking studies. The findings emphasize the need to create and improve AI systems using high-quality data specific to less common languages so that AI tools can be used safely and effectively in dental education and practice.

The primary limitation of this study is that the AI models evaluated may have become outdated by the time of publication owing to intense competition and rapid development in the field, with end-users potentially accessing newer versions. Furthermore, entirely new AI models were introduced to the market during this study. Another important limitation related to prompting is that we compared only the responses generated with and without prompts.

Although the role of prompting was recognized in this study, a detailed analysis comparing different types of prompts (e.g., short vs. detailed, role-based prompts) was not performed. Future research could focus on systematically evaluating how various prompting strategies influence LLM output quality, accuracy, and specificity in the field of REPs.

## 5. Conclusions

While Claude and GPT demonstrated promising accuracy rates for REP-related queries, reaching approximately 78–80% in Eng, particularly when expert prompting was used, their performance is not sufficiently robust for unsupervised clinical application across all languages. On the basis of the findings of this study, the development of LLMs that are specifically fine-tuned for Tr, utilize adaptive tokenization techniques, and are trained on comprehensive, high-quality, domain-specific medical datasets can be recommended. Such targeted model adaptation may help ensure the accuracy, reliability, and clinical relevance of AI-generated answers to medical questions in Tr, thereby supporting safer and more effective integration of AI tools into dental practice. Our findings suggest that although current AI models show promise, further refinement is necessary to ensure safe, reliable, and equitable use in multilingual clinical settings.

The key findings from this study include the following:

The REPs-related queries posed in Eng yielded significantly higher accuracy rates than those posed in Tr.

Grok (Tr nonprompt) delivered the lowest performance scores among the tested models.

Claude (Eng preprompt) achieved the highest average accuracy score.

No AI model demonstrated ≥80% question-answering accuracy.

## Figures and Tables

**Figure 1 healthcare-13-01190-f001:**
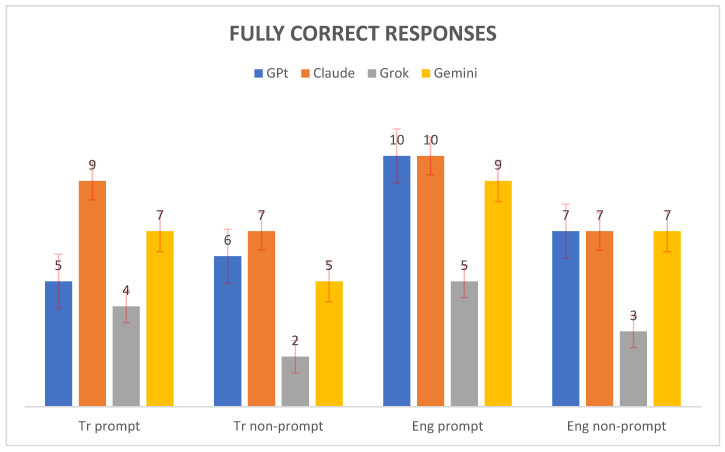
Distribution of correct answers across AI models.

**Figure 2 healthcare-13-01190-f002:**
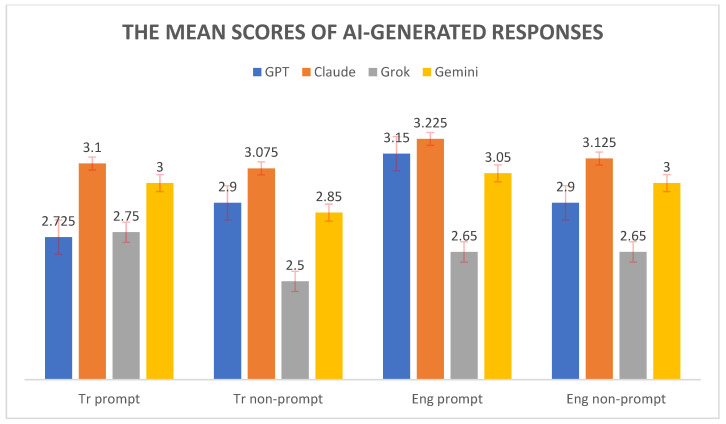
The mean scores of AI-generated responses.

**Table 1 healthcare-13-01190-t001:** Questions designed to ask AI models.

Questions
**A. Indications and Case Selection**
(1) What is the regenerative endodontic procedure applied to immature teeth in the endodontics field of dentistry? What is the ultimate goal?
(2) To which teeth and/or patient groups is the regenerative endodontic procedure applied?
(3) To which teeth and/or patient groups is regenerative endodontic procedure not applied?
(4) Can regenerative endodontic procedures be performed in cases with luxation injuries?
(5) What is recommended for teeth that have undergone regenerative endodontic procedures in patients who are planned for orthodontic treatment?
**B. Preoperative Considerations and Patient Communication**
(6) What information should be given to the patient or legal guardian before a regenerative endodontic procedure is performed?
(7) What are the recommended procedures to be performed at the first appointment in regenerative endodontic procedures?
**C. Clinical Protocol—First and Second Appointments**
(8) What are the recommended procedures to be performed at the first appointment in regenerative endodontic procedures?
(9) Do the AAE, ESE and/or experts have any recommendations regarding the anesthetic solution to be applied in the first session of regenerative endodontic procedures?
(10) How soon after the first appointment should the second appointment be made for regenerative endodontic procedures?
(11) What should be done if the patient’s symptoms have not improved after the first appointment in regenerative endodontic procedures?
(12) What are the second appointment procedure recommendations in regenerative endodontic procedures?
(13) Which anesthetic solution should be used at the second appointment in regenerative endodontic procedures?
**D. Materials and Technical Specifications**
(14) What percentage of ethylene diamine tetra acetic acid (EDTA) solution is recommended in regenerative endodontic procedures?
(15) Which tricalcium silicate cements are recommended in regenerative endodontic procedures? Can you recommend 3?
(16) What should be the recommended thickness of glass ionomer cement to be applied over tricalcium silicate cement in regenerative endodontic procedures?
(17) What is recommended for permanent restoration after regenerative endodontic procedure?
**E. Follow-Up and Outcome Assessment**
(18) How often should patient follow-up be performed after a regenerative endodontic procedure?
(19) Is cone beam computed tomography (CBCT) recommended at follow-up appointments after regenerative endodontic procedures?
(20) What are the success criteria for teeth treated with regenerative endodontics?

**Table 2 healthcare-13-01190-t002:** Descriptive statistics of different AI models, prompts, and languages.

	Total Mean (%)	Std. Deviation
**GPT Tr prompt**	68.13	0.86565
**GPT Tr nonprompt**	72.50	0.82078
**GPT Eng prompt**	78.75	0.85993
**GPT Eng nonprompt**	72.50	0.88258
**Claude Tr prompt**	77.50	0.88258
**Claude Tr nonprompt**	76.88	0.74824
**Claude Eng prompt**	80.63	0.85031
**Claude Eng nonprompt**	78.13	0.74118
**Grok Tr prompt**	68.75	0.83509
**Grok Tr nonprompt**	62.50	0.82717
**Grok Eng prompt**	66.25	0.87509
**Grok Eng nonprompt**	66.25	0.79637
**Gemini Tr prompt**	75.00	0.90321
**Gemini Tr nonprompt**	71.25	0.84449
**Gemini Eng prompt**	76.25	0.88704
**Gemini Eng nonprompt**	75.00	0.76089

**Table 3 healthcare-13-01190-t003:** Repeatability, consistency between experts, temporal consistency, and test–retest reliability coefficients.

	Method	Coefficient	Standard Error	Lower and Upper Limits in the 95% Confidence Interval	
**Repeatability**	**Cohen’s kappa**	0.88	0.03	0.84	0.92
**Consistency between experts**	**Brennan–Prediger**	0.97	0.06	0.85	1.00
	**Fleiss’ kappa**	0.86	0.03	0.80	0.92
	**Krippendorff’s alpha**	0.85	0.01	0.83	0.87
**Temporal consistency and test–retest reliability**	**Intraclass Correlation**	0.92	0.02	0.88	0.96
	**Pearson correlation**	0.89	0.03	0.83	0.95

**Table 4 healthcare-13-01190-t004:** GPT’s language-dependent performance variation: Significant accuracy differences between Tr-Prompted and Eng-Prompted Responses (*p* < 0.05).

	Sum of Squares	df	Mean Square	F	*p* Value	Difference
**Between Groups**	1.834	3	0.611	1.832	0.033 *	GPT Tr prompt < GPT Eng prompt
**Within Groups**	55.888	76	0.735			
**Total**	57.722	79				

* Significant (*p* < 0.05).

**Table 5 healthcare-13-01190-t005:** Results of one-way ANOVA comparing response accuracy across AI models (Grok, Claude) under language-prompting conditions.

		Sum of Squares	df	Mean Square	F	Sig.	Post-Hoc (Tukey HSD)
**Tr nonprompt**	Between Groups	3.484	3	1.161	1.466	0.041 *	Claude > Grok
	Within Groups	49.988	76	0.658			
	Total	53.472	79				
**Eng prompt**	Between Groups	3.934	3	1.311	1.390	0.046 *	Claude > Grok
	Within Groups	57.288	76	0.754			
	Total	61.222	79				
**Tr prompt**	Between Groups	2.059	3	0.686	0.903	0.444	
	Within Groups	57.788	76	0.760			
	Total	59.847	79				
**Eng nonprompt**	Between Groups	2.434	3	0.811	1.277	0.288	
	Within Groups	48.288	76	0.635			
	Total	50.722	79				

* Significant (*p* < 0.05).

## Data Availability

The data presented in this study are available upon reasonable request from the corresponding author.

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
