# Peer review of "The Impact of Language Variability on Artificial Intelligence Performance in Regenerative Endodontics"

_healthcare, 2025, doi:10.3390/healthcare13101190_

Round 1
Reviewer 1 Report
Comments and Suggestions for Authors
This work proposes a study that evaluate the reliability of AI-generated information on REPs, comparing four AI models against clinical guidelines, given AI's increasing role in dentistry. The proposal presents an interesting topic; however, the following aspects were identified:
- In the introduction, it is suggested to provide worldwide statistics on patients requiring the use of REP's, in order to highlight the problems addressed. Also, indicate in greater detail the limitations that dentists currently have in performing REP's, as well as the advantages of having reliable information on REP's that can include text, images or videos. In addition, it is suggested to clearly indicate the scientific contribution(s) of the proposal, with the results obtained. Optionally, it is suggested to include at the end of the introduction a paragraph indicating the structure of the document.
- It is recommended that the first time an acronym is used, its meaning should be described, in this case there are acronyms that were not properly described such as: EDTA, CBCT.
- When a table or graph is presented, not only should it be referred to in the text, but it is also necessary to explain or describe the main data or comparisons presented. In such a way that it is easy for the reader to understand and comprehend. For this reason, it is suggested that the data or results obtained in tables 2, 3, 4 and 5, as well as in figures 1 and 2, be explained.
- It is mentioned that a one-way analysis of variance (ANOVA) was performed which revealed that GTP was statistically significantly more accurate in the responses with Eng indications than in those with Tr indications; however, it is necessary to present a table with the results of the ANOVA of all the AIs analyzed to support the result obtained.
- It is suggested that the authors present their opinion on the future guidelines for the use of AI, particularly in the health sector. In addition, it is necessary that they indicate the future work to be carried out once the results of this work have been obtained.
Author Response
Comments 1: In the introduction, it is suggested to provide worldwide statistics on patients requiring the use of REP's, in order to highlight the problems addressed. Also, indicate in greater detail the limitations that dentists currently have in performing REP's, as well as the advantages of having reliable information on REP's that can include text, images or videos. In addition, it is suggested to clearly indicate the scientific contribution(s) of the proposal, with the results obtained. Optionally, it is suggested to include at the end of the introduction a paragraph indicating the structure of the document.
Response 1: Thank you very much for your valuable suggestions and constructive feedback. In response to your recommendations, we have revised the introduction in the field of patients requiring regenerative endodontic procedures (REPs), provided a more detailed explanation of the current limitations faced by dentists in performing REPs and the advantages of having reliable information on REP's that can include text, images or videos are stated in page 2 line 51-88. Additionally, we have clearly stated the scientific contributions of our study in page 3 line 122-126, page 13 line 472-484 and page 13 line 497-507. We believe these additions have strengthened the context and clarity of our work in line with your comments.
Comments 2: It is recommended that the first time an acronym is used, its meaning should be described, in this case there are acronyms that were not properly described such as: EDTA, CBCT.
Response 2: Thank you for your valuable feedback. I have revised the manuscript to ensure that all acronyms, including EDTA (Ethylenediaminetetraacetic Acid) and CBCT (Cone Beam Computed Tomography), are properly defined at their first mention in the text.
Comments 3: When a table or graph is presented, not only should it be referred to in the text, but it is also necessary to explain or describe the main data or comparisons presented. In such a way that it is easy for the reader to understand and comprehend. For this reason, it is suggested that the data or results obtained in tables 2, 3, 4 and 5, as well as in figures 1 and 2, be explained
Response 3: Thank you very much for your constructive feedback. In response to your suggestion, we have carefully revised the manuscript to ensure that the main findings and comparisons from Tables 2, 3, 4, and 5, as well as Figures 1 and 2, are clearly explained within the text. These additions aim to enhance the clarity and comprehensibility of the results for the reader. We appreciate your valuable input, which has helped us improve the quality and readability of our manuscript.
Table 2 explained in page 6, line 202-204.
Table 3 explained in page 6, line 214-220.
Table 4 explained in page 7, line 227-230.
Table 5 explained in page 7, line 243-251.
Figure 1 explained in page 8, line 263-274.
Figure 2 explained in page 9, line 277-286.
Comments 4: It is mentioned that a one-way analysis of variance (ANOVA) was performed which revealed that GTP was statistically significantly more accurate in the responses with Eng indications than in those with Tr indications; however, it is necessary to present a table with the results of the ANOVA of all the AIs analyzed to support the result obtained.
Response 4: Thank you very much for your valuable feedback. In the revised version of the manuscript, we have added a table summarizing the ANOVA results for Grok and Claude across all prompt conditions. Since there were no statistically significant differences between GPT and the other tested AI models (p > 0.05), and similarly, no significant differences were observed between Claude and Gemini (p > 0.05) or between Grok and Gemini (p > 0.05), these data were not included in Table 5, which contains the ANOVA results. Edits made to Table 5 are marked in red in the main text and indicated on page 8 line 262.
Comments 5: It is suggested that the authors present their opinion on the future guidelines for the use of AI, particularly in the health sector. In addition, it is necessary that they indicate the future work to be carried out once the results of this work have been obtained.
Response 5: Thank you for your insightful comments regarding the inclusion of perspectives on future guidelines for AI use in healthcare and the need to outline future research directions. We have carefully addressed these points in the revised in Discussion section and change can be found page 11 line 391-393, 405-407 and Conclusion section page 13 line 500-507 of our manuscript.
Reviewer 2 Report
Comments and Suggestions for Authors
Dear author,
Why were only these AI models selected? Do these platforms differ technically?
Methodology:
Criteria Selection of 20 questions? Was validation done?
Were the observers trained in the aspect of the knowledge for the question?
For standard answers, were the 20 questions referred to any evidence?
Judging of scoring was done in accordance with?
Author Response
Comments 1: Why were only these AI models selected? Do these platforms differ technically?
Response 1: Thank you for your question regarding the selection of AI models and their technical differences. Please find our response below:
We selected ChatGPT-4o (OpenAI), Claude 3.5 Sonnet (Anthropic), Grok 2 (xAI), and Gemini 2.0 Advanced (Google) because these represent the most advanced, widely used, and technically diverse large language models (LLMs) currently available to the public and professionals. These models are frequently referenced in both academic literature and technology reviews as the leading AI platforms for general and specialized tasks, including healthcare and dentistry. Additionally, a paragraph explaining the technical differences of these platforms and why they were chosen for this study has been added to page 3 line 107-121 in the introduction section.
Comments 2: Criteria Selection of 20 questions? Was validation done?
Response 2: Thank you for your insightful remark regarding the selection and validation of the 20 questions utilized in this study. The 20 questions were meticulously developed based on three authoritative sources in regenerative endodontics: European Society of Endodontology (ESE) Position Statement, American Association of Endodontists (AAE) Clinical Considerations for a Regenerative Procedure, and Expert Consensus on Regenerative Endodontic Procedures published in 2022. These references collectively represent the most up-to-date, evidence-based guidelines in the field and are widely acknowledged by the global endodontic community.
To ensure the relevance and content validity of the questions, each item was initially drafted by the first author and subsequently reviewed by two independent endodontic specialists with clinical and academic expertise in regenerative endodontics. During this review process, each question was evaluated for clarity, clinical applicability, and consistency with the referenced guidelines. Modifications were made based on consensus to ensure that all items accurately reflected the core principles and recommendations outlined in the source documents. The content validity of the question set was ensured through meticulous expert evaluation and its direct correspondence with internationally recognized clinical guidelines.
We have added this comment to the revised manuscript (in Section 2.1) page 3 line 137-141 and marked in red.
Comments 3: Were the observers trained in the aspect of the knowledge for the question?
Response 3: Thank you for raising this important point regarding the qualifications and preparation of the observers involved in the evaluation process. The two observers who independently scored the AI-generated responses are experienced endodontists, each holding academic positions at university dental faculties with extensive clinical and research backgrounds specifically in regenerative endodontics. Both experts are familiar with the current ESE and AAE guidelines. Prior to the evaluation, the observers reviewed the original source documents from which the questions were derived, including the ESE Position Statement, AAE Clinical Considerations, and 2022 expert consensus. This preparatory step ensured a shared baseline understanding of the criteria upon which the AI-generated answers were to be assessed. The very high inter-rater reliability coefficients (e.g., Brennan-Prediger = 0.97, Cohen’s kappa = 0.88, Fleiss’ kappa = 0.86) confirm a consistent and reliable scoring approach between observers. These findings support the validity and rigor of the evaluation process.
We have clarified this aspect in the revised manuscript in page 4 line 147-154, page 5 line 181-185 and marked it in red.
Comments 4: For standard answers, were the 20 questions referred to any evidence?
Response 4: We thank for the valuable comment of the reviewer. Three current guidelines from American Association of Endodontists, European Sociality of Endodontology and 2022 Expert Consensus were referenced for the correct answers to the questions and are stated in the main text. Observers scored the responses of the AI models using these three current guidelines. A paragraph regarding the standardization of answers has been added to the main text on page 5 and line 181-185.
Comments 5: Judging of scoring was done in accordance with?
Response 5: Thank you for your question. The evaluation was conducted using a 4-point holistic scoring rubric method. In this method, the evaluation criteria were applied as follows:
1 = Incorrect (The score the AI receives when its answer is completely wrong)
2 = Partially correct (There are wrong(s) in the AI’s response, but there are also correct answers.)
3 = Correct but incomplete (The AI’s answer is correct but incomplete)
4 = Fully correct (The answer given by the AI is completely correct)
Holistic rubrics typically use a scale with clearly defined performance levels-commonly ranging from 1 to 4-where each level is described by broad qualitative descriptors. This method enables evaluators to make an overall judgment about the quality of the response, emphasizing what the subject can do rather than focusing on deficiencies. For a better understanding of the scoring, page 5 line 187-190 has been added in parentheses and marked in red.
Reviewer 3 Report
Comments and Suggestions for Authors
This study evaluates the reliability of AI-generated information on regenerative endodontic procedures, highlighting the importance of ensuring accurate clinical guidance as AI tools are increasingly integrated into dental practice. The study is highly interesting and presents a very timely and relevant proposal for current clinical and technological contexts.
In the introduction, to better contextualize the topic, the authors could specify why they chose these 4 AIs and what is the prevalence or percentage of use of these tools worldwide. This question is argued because there are other tools, such as SCISPACE, that already provide answers by mentioning the top articles on the topic, which could be more specific to support scientific content.
In the materials and methods, please specify how the 2 endodontic specialists were selected to evaluate the responses, as it is known that even among specialists within the same field, there may be differences in philosophies, level of understanding depending on the length of professional experience, etc.
To avoid confounding bias, it would be interesting for the authors to specify whether any of these platforms have been used on the computer by endodontists. Artificial intelligence is known to be "fed" by information we provide over time.
- What maneuvers were made to ensure that the tools used had not been previously used for similar content examined in this study?
- Do the authors believe that all AIs will behave the same on any computer, regardless of the amount of information fed to those AIs? Explain better.
- Likewise, how did the authors ensure that the 4 AIs had the same level of exposure to the material so as not to interfere with the results?
- How does language structure influence AI comprehension in dental contexts? Does the agglutinative nature of Turkish (and other low-resource, morphologically complex languages) lead to semantic drift or loss of meaning in AI-generated medical content?
- Are certain REP-related terms misinterpreted more often in non-English languages?
Prompt complexity: The authors divided the experimental groups into prompt and non-prompt. This description should be better described in the methodology. In the prompt subgroup, were the commands standardized across all questions and platforms used? This should be better described and specified in the methodology.
- What is the minimum prompt complexity required to optimize clinical performance in LLMs?
- Would simpler prompts like “Explain like a dentist” or “List treatment steps” outperform the used prompt (“Respond as an endodontic specialist”)?
- Can predefined prompt templates improve reliability across different AIs and languages?
In the results, it would be interesting to include information from statistical tests to show in the graphs the existence or not of statistical differences identified by asterisks or (letters, where different letters represent significant statistical differences).
The results can be better organized and structured. From Table 2, the following information from Table 3, Fig. 1, Fig. 2 is already part of the presentation of the results.
The description of the statistical analysis is limited to how the data were collected and organized, how the data were distributed (normal or non-normal), and what statistical tests were used with the statistical program employed and the level of significance.
Some points can be clarified throughout the text:
Limited Insight into Prompting Mechanism
- The role of prompting is recognized, but the paper does not analyze how different prompts impact output (e.g., length, specificity, accuracy).
- Include a prompt engineering breakdown (e.g., compare short vs. detailed prompts, or role-based prompts). This would support or challenge the observed benefit of prompting beyond just numerical outcomes.
Generative models performance under linguistic pressure
- Turkish performance was worse across the board. The authors attribute this to “low-resource language” constraints but do not connect this to generative model architecture or tokenization strategies.
Discuss tokenization bias in GMs trained primarily on English:
- Languages like Turkish lead to longer, fragmented token chains, which may impair coherence.
- Generative models trained mostly on English-based may generate more fluent and accurate content in English, even if translated questions are semantically correct.
- Propose language-specific fine-tuning or adaptive tokenization techniques as improvement paths.
No error or hallucination analysis
- The manuscript lacks a qualitative error analysis of AI responses — which types of misinformation were common? Were they dangerous
- Discuss the implications of the errors/hallucinations in a clinical context (especially for early-career dentists or students).
- Despite Claude and GPT performing relatively well, it is unclear whether they hallucinated outdated or incorrect information.
The authors suggest that “AI is not yet reliable” for REPs, which may be overly broad given that Claude and GPT approached 80% accuracy in some conditions.
Be cautious in phrasing conclusions. Instead of blanket judgments, frame outcomes as context-specific (e.g., “Current models show promise but need refinement for safe, multilingual clinical use”).
How repeatable are AI responses over time — and do changes indicate model drift or improvement? Would re-asking the same REP questions 6 months later yield better, worse, or different answers? How can AI model updates be monitored?
The authors have produced a very interesting article.
Author Response
Comment 1: In the introduction, to better contextualize the topic, the authors could specify why they chose these 4 AIs and what is the prevalence or percentage of use of these tools worldwide. This question is argued because there are other tools, such as SCISPACE, that already provide answers by mentioning the top articles on the topic, which could be more specific to support scientific content.
Response 1: We thank the referee for raising this important point. In response, we have revised the introduction to clarify the rationale behind the selection of GPT-4o, Claude 3.5 Sonnet, Gemini 2.0 Advanced, and Grok 2 for our study in page 3 line 107-111 These LLMs were chosen due to their broad accessibility, wide usage, and general-purpose functionality, which make them more representative of what dental practitioners and students may use in real-world scenarios when seeking information.
The domain-specific tools such as SCISPACE, which are designed to support literature-based scientific inquiry differ significantly from the LLMs evaluated in our study in terms of their user interaction model and response generation. Our focus was on evaluating widely used, open-domain LLMs, which we believe aligns more closely with the information-seeking behaviors relevant to the dental field.
Comment 2: In the materials and methods, please specify how the 2 endodontic specialists were selected to evaluate the responses, as it is known that even among specialists within the same field, there may be differences in philosophies, level of understanding depending on the length of professional experience, etc.
Response 2: Thank you for this insightful comment. We fully agree that evaluative perspectives among specialists may vary depending on clinical philosophy, academic background, and professional experience. To minimize this potential source of heterogeneity, the evaluators were selected based on the following criteria:
Both evaluators were certified endodontic specialists, officially recognized by the Republic of Türkiye Ministry of Health, with over 10 years of combined clinical and academic experience. They are actively involved in postgraduate dental education and possess comprehensive theoretical and practical expertise in regenerative endodontic procedures. Prior to scoring, a calibration session was conducted between the two evaluators to standardize the evaluation criteria based on the guidelines of ESE, AAE, and the 2022 expert consensus document. This process ensured methodological consistency and strengthened inter-rater reliability.
This statement has been added to the Materials and Methods section of the revised article, as suggested, and is marked in red in page 4 line 147-154.
Comments 3:
To avoid confounding bias, it would be interesting for the authors to specify whether any of these platforms have been used on the computer by endodontists. Artificial intelligence is known to be "fed" by information we provide over time.
What maneuvers were made to ensure that the tools used had not been previously used for similar content examined in this study?
Likewise, how did the authors ensure that the 4 AIs had the same level of exposure to the material so as not to interfere with the results?
Response 3: We thank the referee for the comment. One of the inherent strengths of large language models (LLMs) is their ability to learn patterns and refine responses based on interactions with users over time. It is highly likely that, prior to this study, other users posed similar questions regarding regenerative endodontic procedures. However, given the opaque nature of model training and continual learning processes, it is not possible to accurately quantify the extent to which prior user interactions may have influenced the models' responses. Therefore, while acknowledging this as a potential limitation, we attempted to minimize its impact by initiating fresh sessions and posing standardized questions without providing additional context or training material.
Comment 4: Do the authors believe that all AIs will behave the same on any computer, regardless of the amount of information fed to those AIs? Explain better.
Response 4: We thank the referee for the comment. As these AI models operate via cloud-based infrastructures rather than local installations, it was assumed that the responses would remain consistent across different devices, independent of the specific computer used.
Comment 5: How does language structure influence AI comprehension in dental contexts? Does the agglutinative nature of Turkish (and other low-resource, morphologically complex languages) lead to semantic drift or loss of meaning in AI-generated medical content?
Are certain REP-related terms misinterpreted more often in non-English languages?
Response 5: Thank you for your insightful comment regarding the impact of Turkish’s agglutinative morphology on AI comprehension and the potential for semantic drift or loss of meaning in AI-generated medical content.
In our study, the REP-related terms tested in Turkish were directly equivalent to their English counterparts. Terminology such as “regeneration,” “immature,” and “luxation injuries” is commonly used and well-established within Turkish dental and medical literature. During expert evaluation, we did not observe any direct misinterpretation of these specific terms by any of the AI models in Turkish. Therefore, inaccuracies or incomplete answers observed in the Turkish responses were not attributed to misunderstanding of terminology, but rather to factors such as contextual omission or lack of procedural detail.
However, we acknowledge that some clinical concepts—while not incorrectly translated—were sometimes oversimplified or insufficiently elaborated in Turkish, particularly under non-prompted conditions. This reflects not a linguistic mistranslation, but a reduction in conceptual depth, likely due to differences in language modeling capacity for Turkish. We have added a paragraph to the Discussion section to elaborate on this observation, emphasizing the importance of contextual and conceptual accuracy in multilingual clinical AI outputs and marked it in red in page 11 line 394-407
Comment 7: Prompt complexity: The authors divided the experimental groups into prompt and non-prompt. This description should be better described in the methodology. In the prompt subgroup, were the commands standardized across all questions and platforms used? This should be better described and specified in the methodology.
Response 7: Thank you for your valuable comment. In addition to the explanation of prompted and non-prompted subgroups, we agree that standardizing the commands used in the prompted subgroup will provide a better understanding of the methodology. For this reason, the revised text where the standardization is better explained is clarified in the Material methods section on page 5 line 186-190 and marked in red.
Comment 8: In the results, it would be interesting to include information from statistical tests to show in the graphs the existence or not of statistical differences identified by asterisks or (letters, where different letters represent significant statistical differences).
Response 8: thank you for your contribution. The additions have been made to the tables and marked in red in the revised text.
Comment 9: The results can be better organized and structured. From Table 2, the following information from Table 3, Fig. 1, Fig. 2 is already part of the presentation of the results.
Response 9: Thank you very much for your constructive suggestion regarding the organization of the results section. In response, we have removed the mean score column from Table 2, as this information is already effectively presented in Figure 2. Additionally, since the standard error of the mean (SEM) is visually indicated by the error bars in the figure 2, we have also omitted the SEM values from the table to avoid redundancy. We believe that these adjustments streamline the presentation and make it easier for readers to interpret the key findings with greater clarity.
Comment 10: The description of the statistical analysis is limited to how the data were collected and organized, how the data were distributed (normal or non-normal), and what statistical tests were used with the statistical program employed and the level of significance.
Response 11: Thank you very much for your valuable feedback. I appreciate your suggestion regarding the description of the statistical analysis. In the revised version, we clearly state how the data were distributed (stated statistical analysis section page 5-6 line 199-202). We also provide detailed information about the statistical tests, including the statistical software and the significance level used (stated page 5, line 192-194). We also explain the reasons and explanations of the statistical tests performed on page 6, line 206-212 and mark them in red.
Comments 12: Limited Insight into Prompting Mechanism
- The role of prompting is recognized, but the paper does not analyze how different prompts impact output (e.g., length, specificity, accuracy).
- Include a prompt engineering breakdown (e.g., compare short vs. detailed prompts, or role-based prompts). This would support or challenge the observed benefit of prompting beyond just numerical outcomes.
- What is the minimum prompt complexity required to optimize clinical performance in LLMs?
- Would simpler prompts like “Explain like a dentist” or “List treatment steps” outperform the used prompt (“Respond as an endodontic specialist”)?
- Can predefined prompt templates improve reliability across different AIs and languages?
Response 12: We would like to sincerely thank the referee for this valuable comment regarding the role of prompting and the suggestion to analyze how different types of prompts (e.g., short vs. detailed, role-based prompts) affect the outputs. We agree that a deeper evaluation of the prompting mechanisms would provide important additional insights into LLM performance.
However, we believe that such an analysis would extend beyond the scope and primary aim of the current study, which was focused on evaluating LLMs’ capabilities specifically in the field of regenerative endodontics. Considering the expertise and interests of the target audience, we feel that a detailed breakdown of prompt engineering strategies may not align with the main objectives of this paper.
Nevertheless, we have acknowledged this important point in the discussion section page 13 line 491-495 by suggesting that future studies could explore the effects of different prompting strategies on the performance of LLMs in dental applications.
The competition among LLM developers is intense, and new versions of AI models are released at a rapid pace. For instance, although we tested Claude 3.5 Sonnet during the study period, a more advanced version (Claude 3.7) has already been launched. Additionally, due to the continual learning mechanisms embedded in these systems, even the same model may evolve and improve over time. Conducting a comprehensive analysis of the prompt engineering mechanism (e.g., comparing short, detailed, or role-based prompts) would require re-running the entire study under standardized conditions to ensure a fair comparison. Unfortunately, due to resource constraints, it was not feasible to undertake such an extensive re-evaluation. However, we acknowledge this as a limitation of our study and recommend that future research address the impact of different prompting strategies more systematically.
Comments 13: Generative models performance under linguistic pressure
- Turkish performance was worse across the board. The authors attribute this to “low-resource language” constraints but do not connect this to generative model architecture or tokenization strategies.
- Discuss tokenization bias in GMs trained primarily on English:
- Languages like Turkish lead to longer, fragmented token chains, which may impair coherence.
- Generative models trained mostly on English-based may generate more fluent and accurate content in English, even if translated questions are semantically correct.
- Propose language-specific fine-tuning or adaptive tokenization techniques as improvement paths.
Response 13: Thank you for raising this important point regarding the relationship between generative model architecture, tokenization strategies, and the observed lower performance of AI models in Turkish for REP-related queries.
We have expanded the discussion section page 10 line 343-352 in the revised manuscript added a paragraph to discussion includes limitations due to the Turkish complex morphological characteristics. And we also proposed in the conclusion section page 13 line 500-507, the development of LLMs that are specifically fine-tuned for Turkish and trained on com-prehensive, high-quality, domain-specific medical datasets.
Comments 14: No error or hallucination analysis
- The manuscript lacks a qualitative error analysis of AI responses — which types of misinformation were common? Were they dangerous
- Discuss the implications of the errors/hallucinations in a clinical context (especially for early-career dentists or students).
- Despite Claude and GPT performing relatively well, it is unclear whether they hallucinated outdated or incorrect information.
Response 14: Thank you for your valuable comments. We agree with you that AI responses should be evaluated not only in terms of their quantitative accuracy rates but also in terms of their content quality and risks in the clinical context. In this context, a comprehensive qualitative error and hallucination analysis has been added to the Discussion section of our study. In this analysis, errors observed in AI responses were thematically divided into three groups according to their content and clinical risk levels:
(i) Minor errors of low theoretical significance: Content that does not directly affect clinical practice, such as definitional ambiguities or incomplete term use.
(ii) Moderate incorrect or missing information: Deficiencies that may indirectly affect clinical protocols, such as anesthetic preference, irrigant order, or material use.
(iii) Clinically important and contrary to guidelines recommendations: For example, Claude and GPT models occasionally recommend the use of CHX, which is contraindicated in REPs due to its cytotoxic effects on apical tissues. Similarly, GPT's recommendation of using NaOCl in the second session also contradicts the guidelines.
The potential effects of these errors in the clinical context can be misleading, especially for dentists and students at the beginning of their careers. The fact that AI appears to be based on scientific references increases the potential to mislead the user when it contains incorrect information. Users with limited clinical experience may accept these contents as correct and perform practices contrary to the protocol. This both reduces treatment success and increases the risk of harming the patient.
Although Claude and GPT models have relatively higher accuracy rates, it has been observed that they sometimes provide hallucinations and outdated information. This is clearly seen in recommendations such as CHX or post placement. Therefore, high performance does not mean error-free. Thanks to these additions, our study has become a detailed analysis of AI models not only in terms of performance comparison but also in terms of content reliability. We thank you for filling an important gap thanks to your warning.
An explanation of this issue has been added to the Discussion section and is highlighted in red in page 11 line 376-393.
Comments 15: The authors suggest that “AI is not yet reliable” for REPs, which may be overly broad given that Claude and GPT approached 80% accuracy in some conditions.
Be cautious in phrasing conclusions. Instead of blanket judgments, frame outcomes as context-specific (e.g., “Current models show promise but need refinement for safe, multilingual clinical use”).
Response 15: Thank you very much for your valuable feedback regarding the phrasing of our conclusions. In line with your suggestion, we have carefully revised the Conclusion section (in page 13, line 497-507) to avoid overly broad statements and to more accurately reflect the context-dependent performance of the evaluated AI models. We believe this revision provides a more balanced and precise summary of our findings and better aligns with the nuanced results presented in the manuscript. Thank you again for your constructive guidance.
Comment 16: How repeatable are AI responses over time — and do changes indicate model drift or improvement? Would re-asking the same REP questions 6 months later yield better, worse, or different answers? How can AI model updates be monitored?
Response 16: Thank you for your valuable comment regarding the repeatability of AI responses over time and the implications of model drift or improvement.
In our study, we assessed the repeatability of AI-generated answers by posing the same REP-related questions to each model at 10-day intervals. The results showed that the AI answers were very consistent over short periods, supported by strong statistical measures (like Cohen’s kappa = 0.88 and Intraclass Correlation Coefficient = 0.92; see Table 2).
The competition among LLM developers is intense, and new versions of AI models are released at a rapid pace. For instance, Claude 3.5 Sonnet, which we tested during the study period, has already given way to a more advanced version, Claude 3.7. Additionally, due to the continual learning mechanisms embedded in these systems, even the same model may evolve and improve over time. Since the AI models tested in this study will have been updated after 6 months, it will theoretically not be possible to ask the same questions to the same AI model after six months. If the same REP questions were re-asked after six months, it is likely that the answers would differ to some extent. These changes may reflect improvements—such as increased accuracy, more up-to-date information, or enhanced reasoning-or, conversely, may introduce new inconsistencies or errors depending on the nature and quality of the model updates. This phenomenon, known as model drift, is well recognized in the AI field and underscores the importance of ongoing performance monitoring.
In summary, although current AI models show high short-term repeatability, their responses may change over time due to updates, retraining, and model drift. Therefore, the reliability and clinical safety of AI-generated information in dentistry should continue to be tested for monitoring updates.
Comment17: The authors have produced a very interesting article.
Response 17: Thank you very much
Reviewer 4 Report
Comments and Suggestions for Authors
This is an important contribution to the growing field of AI in clinical dentistry and endodontics. With the incorporation of the above refinements especially around prompting clarity, model context, and data visualization this work will be well-positioned for publication in Healthcare. Below are my detailed comments and suggestions aimed at enhancing the clarity, depth, and overall impact of your manuscript:
- While you mention that both prompted and non-prompted conditions were tested, please specify the exact wording used for the prompt. This detail will improve reproducibility and clarity for future researchers.
- The description of AI models (GPT, Claude, Grok, Gemini) could be slightly expanded to include their architectural differences or access limitations. Consider adding a small table summarizing each model's known strengths and limitations.
- Figures 1 and 2 are informative, but the inclusion of 95% confidence intervals or error bars for the scores would help the reader interpret statistical significance visually.
- You show a statistically significant drop in GPT performance for Turkish vs. English prompts. It would be valuable to explore why this might be the case - for example, due to dataset sparsity or morphological complexity of Turkish.
- Consider categorizing the 20 questions into themes (e.g., Indications, Procedure Steps, Materials, Follow-Up), which may help in understanding which areas models perform best/worst in.
- Since Gemini's newer version performed much better than previous evaluations, it would be helpful to briefly mention the model version dates for all four AI systems.
- Consider including a short paragraph addressing AI hallucinations in clinical settings. This could emphasize the importance of human verification in the current state of AI adoption.
Comments on the Quality of English Language
I recommend a light professional English editing pass focused on grammar, clarity, and stylistic consistency. This will ensure smoother flow and polish the manuscript for publication quality without altering the scientific content.
Author Response
|
Comments 1: While you mention that both prompted and non-prompted conditions were tested, please specify the exact wording used for the prompt. This detail will improve reproducibility and clarity for future researchers.
Response 1: Thank you very much for your valuable comment. We agree that providing the exact wording of the prompt will enhance the reproducibility and clarity of our study. Accordingly, we have added the precise wording used in the prompted condition to the Materials and Method “Generating Answers” section of the manuscript. The sentence now reads in page 5, line 166-171
Comments 2: The description of AI models (GPT, Claude, Grok, Gemini) could be slightly expanded to include their architectural differences or access limitations. Consider adding a small table summarizing each model's known strengths and limitations.
Response 2: Thank you for your valuable suggestion. In response, we have expanded the Introduction to provide concise descriptions of each model’s core architecture and, unique features and can be found in page 3 line 111-121. And also in the Discussion section, expanded to include their architectural differences or access limitations stated page 10 line 321-339. We believe that this expanded narrative in the Introduction and Discussion offers readers a clear and accessible overview of the main differences between the models, in line with your recommendation, while maintaining the flow and conciseness of the manuscript. Should you consider a summary table essential, we are happy to include one in a subsequent revision.
Comments 3: Figures 1 and 2 are informative, but the inclusion of 95% confidence intervals or error bars for the scores would help the reader interpret statistical significance visually.
Response 3: We appreciate the reviewer's suggestion to include 95% confidence intervals or error bars in Figures 1 and 2 to enhance the visual interpretation of statistical significance. We have now revised both figures to incorporate error bars representing the 95% confidence intervals for all reported scores.
Comments 4: You show a statistically significant drop in GPT performance for Turkish vs. English prompts. It would be valuable to explore why this might be the case - for example, due to dataset sparsity or morphological complexity of Turkish.
|
|
Response 4: Thank you very much for your contribution. The following paragraph has been added to the discussion in line with your suggestions. The paragraph now reads in discussion section page 10, line 343-352 and 371-373.
|
Comments 5: Consider categorizing the 20 questions into themes (e.g., Indications, Procedure Steps, Materials, Follow-Up), which may help in understanding which areas models perform best/worst in.
Response 5: Thank you for your valuable suggestion regarding the thematic categorization of the questions. We agree that grouping the questions into clinical domains can enhance interpretability and provide clearer insights into the strengths and limitations of AI model performance. Accordingly, we have categorized the 20 questions into four primary themes based on clinical relevance:
- Indications and Case Selection
- Preoperative Considerations and Patient Communication
- Clinical Protocol – First and Second Appointments
- Materials and Technical Specifications
- Follow-Up and Outcome Assessment
This thematic structure has been integrated into the revised manuscript (Table 1) to facilitate a more nuanced evaluation of model accuracy within specific knowledge domains and marked in red. table now reads in page 4-5.
Comments 6: Since Gemini's newer version performed much better than previous evaluations, it would be helpful to briefly mention the model version dates for all four AI systems.
Response 6: Thank you for your valuable suggestion. We agree with your comment regarding the importance of specifying the model version dates for all four AI systems evaluated in our study. We have added this information to the manuscript in page 10 line 318-334 to provide better context for our comparative analysis.
Comments 7: Consider including a short paragraph addressing AI hallucinations in clinical settings. This could emphasize the importance of human verification in the current state of AI adoption.
Response 7: Thank you for your valuable comment regarding the inclusion of a discussion on AI hallucinations in clinical settings and the importance of human verification.
In our revised manuscript, we have addressed this issue in the Discussion section. Specifically, in the Discussion page 9 line 302-311 and page 11 line 376-393 we emphasize that AI-generated content-even when it appears authoritative-may include fabricated or guideline-inconsistent information.
In line with your suggestion, we have further clarified in the revised manuscript that human oversight and expert validation are essential whenever AI is used in clinical dentistry, especially given the current limitations and risks of hallucinations.
Thank you again for your constructive feedback, which has helped us strengthen the manuscript’s clinical relevance and cautionary guidance.
Round 2
Reviewer 1 Report
Comments and Suggestions for Authors
This work proposes a study that evaluate the reliability of AI-generated information on REPs, comparing four AI models against clinical guidelines, given AI's increasing role in dentistry. The proposal presents an interesting topic; however, the following aspects were identified:
1.- Although the contribution of the proposal is noted, it is suggested that the authors explicitly indicate “The scientific contribution of this work is ....” so that it can be easily identified.
2. Verify the citation style required by the journal.
Author Response
Reviewer 1 Round 2
Dear Reviewer,
We would like to express our sincere gratitude for your thoughtful review and constructive comments regarding our manuscript entitled “The Impact of Language Variability on AI Performance in Regenerative Endodontics.” We appreciate the opportunity to address your suggestions and have revised our manuscript accordingly. Please find our responses to each point below:
Reviewer Comment 1:
“Although the contribution of the proposal is noted, it is suggested that the authors explicitly indicate ‘The scientific contribution of this work is ....’ so that it can be easily identified.”
Response 1:
Thank you for highlighting the importance of clearly stating the scientific contribution. In response, we have added explicit statement to the discussion section page 13 line 480-491. We believe this addition makes the scientific contribution of our work clear and easily identifiable for readers.
Reviewer Comment 2:
“Verify the citation style required by the journal.”
Response 2:
We have carefully reviewed the journal’s guidelines for citation and referencing style. All in-text citations and the reference list have been revised to strictly adhere to the required format. We have double-checked that all references are consistently formatted and that every cited work is included in the reference list.
We thank you again for your valuable feedback, which has helped us to improve the clarity and quality of our manuscript. We hope that our revisions satisfactorily address your comments.